# Photonic Crystal Surface Modes for Trapping and Waveguiding of Ultracold Atoms

**DOI:** 10.3390/s23218812

**Published:** 2023-10-30

**Authors:** Valery Konopsky

**Affiliations:** Institute of Spectroscopy, Fizicheskaya, 5, Troitsk, Moscow 108840, Russia; konopsky@isan.troitsk.ru

**Keywords:** atom-based quantum sensors, atom interferometry on a chip, optical surface modes, photonic crystal waveguides, optical Bloch surface waves

## Abstract

The design of a photonic system for the trapping and waveguiding of ultracold atoms far above a dielectric surface is proposed and analyzed. The system consists of an optical rib waveguide deposited on a planar one-dimensional photonic crystal, which sustains two wavelengths of photonic crystal surface modes tuned in the red and blue sides relative to the atomic transition of the neutral atom. The addition of a third blue-tuned wavelength to the system allows the neutral atoms to be stabilized in the lateral dimension above the rib waveguide. Trapping atoms at relatively large distances, more than 600 nm above the dielectric surface, allows to reduce the influence of Casimir–Polder forces in this system. The detailed design methodology and specifications of the photonic system are provided. The presented design can be employed in atomic chips and quantum sensors.

## 1. Introduction

The development of quantum sensors [1] based on coherent atomic circuits is important for the realization of many potential technological applications [2], such as gravimeters [3,4,5,6,7], gradiometers [8] and gyroscopes [9,10]. Atom chips [11,12,13] provide a versatile technique for the generation and coherent manipulation of ultracold atoms for such sensors on the micrometer scale.

For the successful development of compact and robust atom chips integrated with guided atomic interferometers, the optimal choice of the appropriate interface for the light–atom interaction is of crucial importance. In this study, we examine the trapping and waveguiding of ultracold atoms employing two-color optical surface waves propagating along the external boundary of a rib waveguide located on top of a specially designed planar one-dimensional photonic crystal (1D PC).

The use of optical waveguides to guide atoms was proposed in 1993 by Letokhov et al. [14]. They proposed to create a potential well inside a hollow core fiber by a laser with a red-detuned light directed into the fiber. Since then, the use of optical waveguides has been identified as a key element of atomic chip interferometry, and many types of optical waveguides have been considered as candidates for the best interfaces of light–atom interactions within them. The employment of integrated optical waveguides to trap and guide ultracold atoms above planar surfaces has been analyzed in [15,16]. They exploit an earlier proposal by Ovchinnikov et al. [17] to use two colors of light, with red and blue detuning and different evanescent decay lengths, to obtain a planar (one-dimensional) trap with a potential minimum above a dielectric prism. That is, the insertion of optical waveguides into a planar surface should add lateral confinement and restrict the ensemble of cold atoms in two dimensions.

The main problem here is that the evanescent decay length into an external medium in such waveguides cannot be very large, since the effective refractive index of an optical wave $n_{\mathrm{eff}}$ in a standard waveguide cannot be smaller than the refractive index of the waveguide substrate $n_{0}$. The penetration length of the evanescent wave intensity (i.e., $I={\left|E\right|}^{2}$) in the external medium is
(1)$$L_{e}=\frac{\lambda }{4\pi \sqrt{\rho^{2}-n_{e}^{2}}}\;.$$ Here, ρ is either the numerical aperture of the light beam in the prism, $\rho =n_{0}\sin\left(\theta_{0}\right)$, or the effective refractive index of the optical mode in the corresponding planar 1D slab waveguide, $\rho =n_{\mathrm{eff}}\left(w_{\infty }\right)$. The refractive index of the external medium (i.e., a vacuum for atom optics applications) is $n_{e}=1$. The value of $L_{e}$ can only be very large in the case when the difference $\left(\rho^{2}-1\right)$ is small.

It is shown below that this condition $\rho =n_{\mathrm{eff}}\left(w_{\infty }\right)≃1$ can be satisfied for a photonic crystal (PC) waveguide, in contrast to a standard optical waveguide. It is this unique property of PC waveguides that makes it possible to design the photonic system that sustains the ultracold atomic ensembles at relatively large distances above the dielectric surface. This is the reason why, in this communication, we propose and analyze such an innovative waveguide structure, based on planar 1D PC, for ultracold atom trapping and waveguiding.

## 2. Materials and Methods

### 2.1. Planar Photonic Crystal Waveguides and Their Differences from Standard Waveguides

The effective index of a two-dimensional optical waveguide $n_{\mathrm{eff}}\left(w\right)$ with finite lateral width w (i.e., rib or ion-implanted waveguide) is somewhat smaller than the effective index of the corresponding planar 1D slab waveguide $n_{\mathrm{eff}}\left(w_{\infty }\right)$, where the width of w=∞ (i.e., extends to infinity in the lateral direction). This diminution of $n_{\mathrm{eff}}\left(w\right)$ is due to the zigzag propagation (from left to right) of the wave in the plane of the waveguide (in the ray model approximation), resulting in a reduction in the projection of the wavevector in the z-direction. But the penetration length of the waveguide mode intensity into the external medium for a two-dimensional waveguide is still determined by the Formula (Equation 1), where the effective index of the corresponding planar 1D slab waveguide $n_{\mathrm{eff}}\left(w_{\infty }\right)$ is inserted as ρ.

According to Equation (Equation 1), only modes with $n_{\mathrm{eff}}\left(w_{\infty }\right)≃1$ can have large penetration length in the medium with $n_{e}=1$. In standard waveguides, the optical field confinement in the vertical direction is the result of total internal reflection (TIR) on both sides of the waveguide layer. As a result, $n_{\mathrm{eff}}\left(w_{\infty }\right)>n_{0}⩾n_{e}$ (i.e., $n_{\mathrm{eff}}\left(w_{\infty }\right)>1$), and the maximum possible penetration length cannot be greater than $L_{e}\left(n_{0}\right)=\lambda /\left(4\pi {\left[n_{0}^{2}-n_{e}^{2}\right]}^{1/2}\right)$. For a waveguide on a quartz substrate with $n_{0}=1.46$ and λ=850 nm, this gives $L_{e}\left(1.46\right)⩽64$ nm. In [15], the authors consider sodium fluoride (the lowest-index common optical mineral, $n_{0}=1.32$) as a possible substrate, but it yields only $L_{e}\left(1.32\right)⩽79$ nm.

One possible way to mitigate this limitation is to use a freely suspended thin dielectric film [18,19] to achieve $n_{0}=n_{e}=1$. With this approach, the following penetration length values for the *intensity* of evanescent waves were obtained [18]: $L_{e}\left(1.0\right)=87.5$ nm and 69.75 nm for modes excited in a 300 nm thick suspended silica film with waveguide rib height h=15 nm and width w=2μm (at wavelengths λ=850 nm and 720 nm, respectively).

This communication presents a different approach to overcome this limitation, in which the uniform waveguide substrate is replaced by a planar 1D photonic crystal, which is designed to create a photonic bandgap in the spectral region of interest. Thus, in this case, the optical field confinement in the vertical direction is due to reflection from the photonic bandgap on one side (bottom) and total internal reflection—as usual—on the another side (top) of the waveguide, as shown in Figure 1.

The planar 1D PC is a simple dielectric stack. It is durable and compatible with existing technologies for producing dielectric mirrors; so, its fabrication is much easier than suspended thin dielectric films. In the vicinity of the interface between the planar 1D PC and the external medium, optical surface waves can be excited subject to an appropriate choice of the thicknesses of the double layers and the thickness of the final truncated layer [20].

These optical surface waves are excitations of photonic crystal surface modes (PC SMs), also known as ‘(asymmetric) planar Bragg waveguide modes’, ‘surface waves in periodic layered media’, ‘photonic bandgap surface modes’, ‘optical Bloch surface waves’, ‘photonic crystal surface waves’ and ‘surface waves in multilayer coatings’. They were first studied both theoretically [21,22] and experimentally [23] in the 1970s. Twenty years later, the excitation of optical surface modes in the Kretschmann-like configuration was demonstrated [24,25]. In recent years, PC SMs have found an increasing number of applications in the fields of optical sensors [26,27,28,29,30] and optical biosensors [31,32,33,34,35,36,37], as well as other fields [38,39,40,41,42,43].

The penetration length in the external medium can be very large for these modes, as their effective refractive index can be infinitesimally close to the refractive index (RI) of the external medium. This is a unique property of PC SMs that makes their excitation with $\rho =n_{\mathrm{eff}}\left(w_{\infty }\right)≃1$ quite possible. In the first optical biosensor based on these modes [31], this property was already used to distinguish between a surface adsorption and bulk RI (i.e., RI far from the surface).

### 2.2. Design Methodology for a 2D Waveguide Placed on a 1D PC

#### 2.2.1. Design of a 1D PC

The design of a 1D PC, sustaining two PC SMs at wavelengths $\lambda_{1}=850$ nm and $\lambda_{2}=640$ nm, was performed using a free Windows program available at [44]. Results are presented in Figure 2. The wavelengths are chosen to be detuned far enough away from the $D_{1}$, $D_{2}$ transitions of the ${}^{87}$Rb atom ($\lambda_{D_{1}}=795$ nm, $\lambda_{D_{2}}=780$ nm) on the blue and the red sides. The light potential (which causes the attraction and repulsion forces) scales as I/Δ, while the light scattering rate (which causes the heating rate) scales as $I/\Delta^{2}$ [45], where Δ is a detuning from the *D* transitions (see Equations (Equation 3) and (Equation 4) below). Therefore, to increase the trap lifetime, it is better to be away from the resonance if the intensities still allow the desired trap depth to be reached. The larger detuning in the blue side is due to the expectation that a larger value of intensity of the repulsive (blue-detuned) light beam will be needed to form a well trap potential of the desired shape.

The following 1D PC structure is designed to sustain both $\lambda_{1}=850$ nm and $\lambda_{2}=640$ nm PC SMs:(2)$$\mathrm{prism}/{\left(\mathrm{LH}\right)}^{N}L^{\prime \prime }{\mathrm{HL}}^{\prime }/\mathrm{vacuum}\;,$$
where *L* is the $SiO_{2}$ layer (thickness $d_{1}=170.73$ nm), *H* is the $TiO_{2}$ layer (thickness $d_{2}=86.46$ nm), $L^{\prime \prime }$ is the extended $SiO_{2}$ layer (thickness $d_{12}=304$ nm) and $L^{\prime }$ is the final $SiO_{2}$ layer (thickness $d_{3}=207$ nm). The upper part of Figure 2C shows the embedded representation of the structural Formula (Equation 2). The RIs of $SiO_{2}$, $TiO_{2}$ layers used in the calculations are given in Table 1. The obtained parameters of PC SMs in this planar 1D PC structure are also presented in Table 1 and in Figure 2.

It can be seen that this 1D PC is designed so that both modes at $d_{3}=207$ nm have an excitation angle very close to the TIR angle, and hence, the effective RI is close to unity ($\rho =n_{\mathrm{eff}}\left(w_{\infty }\right)≃1$). As a consequence, the penetration lengths of their intensities into the vacuum are very long (see $L_{e}$ in Figure 2C,D and in Table 1).

#### 2.2.2. From 1D PC Waveguide to 2D Waveguide

Note that if we reduce the thickness of the final $SiO_{2}$ layer by 25 nm (or more), from $d_{3}=207$ nm to 182 nm, then no PC SMs will be excited on such planar 1D PC near ρ≃1 (up to ρ=1.17 for $\lambda_{2}=640$ nm); see Figure 2B. Therefore, if we use the planar 1D PC with $d_{3}=182$ nm and a waveguide of height h=25 nm above it, the excitation of the PC SMs will take place inside the two-dimensional waveguide only, subject to its width w being sufficient. Thus, if excitation is performed by beams with excitation angles close to $\theta_{0}$, then only PC SMs with $\rho =n_{\mathrm{eff}}\left(w_{\infty }\right)=n_{0}\sin\left(\theta_{0}\right)$ will be excited, i.e., only modes in two-dimensional waveguides of width w but not modes in the surrounding planar slab waveguide. Moreover, the penetration of such a waveguide wave into the external medium (vacuum in our case) will be the same as for the wave in the planar slab waveguide in Figure 2A,C,D, where the waveguide width $w_{\infty }$ is extended to infinity in the lateral direction. The Equation (Equation 1) still gives the correct value of $L_{e}$ when $\rho =n_{\mathrm{eff}}\left(w_{\infty }\right)$ is substituted. This is because the same wave propagates in the 2D waveguide as in the planar slab 1D waveguide but in a zigzag fashion from left to right, if considered in the ray model approximation.

#### 2.2.3. Design of a 2D Waveguide Placed on a 1D PC

The mode analysis of the 2D waveguide on the surface of the 1D PC was performed by using both a semianalytical mode solver based on the Variational Effective Index Method (VEIM) [46,47] and a numerical Finite Difference Eigenmode (FDE) solver [48] with the Lumerical Inc. (Vancouver, BC, Canada) MODE software (2023 R1, Version 7.21.3262). Both approaches gave very similar results for effective refractive indices and mode profiles.

### 2.3. Calculating the Optical Dipole Potential at Large Wavelength Detuning

For very large detunings that significantly exceed the splitting of the D-line doublet, the following approximate expression for the dipole potential can be used [45]:(3)$$U_{\mathrm{dip}}\left(x,y,\omega \right)=\frac{3\pi c^{2}}{2\omega_{D}^{3}}\frac{\gamma_{D}}{\left(\omega -\omega_{D}\right)}I\left(x,y\right)\;,$$
where $\omega_{D}=2\pi \times 380.77$ THz is the frequency of the center of the D-line doublet, and $\gamma_{D}=37$ MHz is the mean value of natural linewidths of the transitions $D_{1}$, $D_{2}$. Using the two-dimensional spatial distributions of light intensities I(x,y) shown in Figure 3, we obtain the well trap potential shown in Figure 4 (in temperature units $U/k_{B}$). Atomic light scattering rate also may be calculated in this approximation:(4)$$\hbar \;\Gamma_{\mathrm{sc}}\left(x,y,\omega \right)=\frac{\gamma_{D}}{\left(\omega -\omega_{D}\right)}U_{\mathrm{dip}}\left(x,y,\omega \right)\;.$$Each scattered photon imparts a recoil energy $E_{\mathrm{rec}}={\left(\hbar \omega /c\right)}^{2}/\left(2m\right)$ to the atom of mass *m*. As a result, taking into account the approximated expressions (Equation 3) and (Equation 4), a trap lifetime τ can be estimated as
(5)$$\frac{1}{\tau }≃\frac{2E_{\mathrm{rec}}}{U_{\mathrm{dip}}}\Gamma_{\mathrm{sc}}=\frac{2E_{\mathrm{rec}}}{\hbar |\omega -\omega_{D}|}\gamma_{D}\;.$$In addition to atom losses due to photon scattering, the trap lifetime can be reduced by background gas collision losses, which outweigh photon scattering losses at large detuning values [49].

The total potential of the atom $U_{\mathrm{total}}\left(x,y\right)$ near the surface is the sum of the optical potentials $U_{\mathrm{dip}}\left(x,y,\omega \right)$ of all participating light beams and the Casimir–Polder potential together with the gravitational potential:(6)$$U_{\mathrm{total}}\left(x,y\right)=\sum_{\omega =\omega_{1}}^{\omega_{3}}U_{\mathrm{dip}}\left(x,y,\omega \right)-\frac{C_{3}\lambda_{\mathrm{eff}}}{x^{3}\left(x+\lambda_{\mathrm{eff}}\right)}-mgx\;,$$
where $C_{3}=5.7\times 10^{-49}$ J m^3^ is the van der Waals coefficient, $\lambda_{\mathrm{eff}}=710/\left(2\pi \right)$ nm is the reduced wavelength [18,50] describing the attractive interaction between atoms and the glass surface and g=9.8 m/s^2^ is the acceleration of free fall.

## 3. Results

### 3.1. Example of Specification of a 2D Waveguide Placed on a 1D PC

After selecting a waveguide height of 25 nm (above the last layer $L^{\prime }$ with a thickness of $d_{3}=182$ nm, see Figure 2), the width of the waveguide has to be selected. Both the semianalytical VEIM solver and the numerical FDE solver show that the waveguide with w=3μm sustains single TE00 mode propagation for $\lambda_{1}=850$ nm and $\lambda_{2}=640$ nm with effective RI $n_{{\mathrm{eff}}_{1}}=1.0012$ and $n_{{\mathrm{eff}}_{2}}=1.0074$, respectively. In addition, at wavelength $\lambda_{3}=633$ nm, the TE01 mode with $n_{{\mathrm{eff}}_{3}}=1.0076$ is also supported in the waveguide with 3μm width. The next (TE00) mode at $\lambda_{3}=633$ nm has an effective index of $n_{{\mathrm{eff}}_{30}}=1.0163$. Therefore, only three modes with the electric field intensity distribution shown in Figure 3 are excited at excitation angles of ρ⩽1.016. The third mode with $\lambda_{3}=633$ nm is needed to provide ’lateral support’ for the ultracold atoms in the optical trap above the waveguide with 3μm width.

The calculated mode parameters for the rib waveguide with w=3μm,h=25nm are presented in Table 2. Their effective RI, the fraction of Poynting energy transmitted through the vacuum, the effective area of the modes and the losses are given. The optical power in these modes required to form a 100 μK trap, as in Figure 4A–C, is also indicated. Note that in these waveguides, a significant fraction of the Poynting energy $S_{z}$ propagates through the external medium (vacuum). This is a unique property of the presented 1D PC waveguides that is unattainable for standard waveguides, where most of the Poynting energy is transmitted through the waveguide core and only a small fraction of the evanescent wave is available to manipulate atoms in the vacuum.

The mode losses were calculated assuming that the imaginary part of the RI of $SiO_{2}$ and $TiO_{2}$ is $10^{-5}$. According to [51], values as low as $Im\left(n_{j}\right)<10^{-5}$ in this spectral range can be obtained for these dielectric materials using ion-assisted electron beam deposition. We set $Im\left(n_{j}\right)=10^{-5}$ to account for scattering losses, which can be quite prominent for PC SMs with effective refractive indices close to unity.

### 3.2. Optical Trap above the Waveguide Located on the 1D PC

The total well trap potentials of the atom above the rib glass waveguide on 1D PC, which are calculated using the Equation (Equation 6), are shown in Figure 4. Figure 4A shows a potential with two wavelengths: attracting (red-detuned, $\lambda_{1}=850$ nm) and repulsive (blue-detuned, $\lambda_{2}=640$ nm). One can see in this figure a common problem for this type of potential: weak localization in the lateral dimensions and a decrease in the height of the well trap near the edge of the waveguide (i.e., near y=±1.5μm in our case). In the vicinity of these points, atoms will escape from the trap. To solve this problem, the third beam with the wavelength of $\lambda_{3}=633$ nm (TE01) is added, and the resulting total potential is shown in Figure 4B. That is, all the modes shown in Figure 3 are used in Figure 4B. It is clear that such ’lateral support’ dramatically improves lateral localization and eliminates atom escape paths to the surface through points y=±1.5μm.

In Figure 4C, a vertical cross-section of the potential well at y=0 is shown. The minimum of the well is at h=665 nm from the surface. This should provide a very stable confinement of atoms at large distances from the dielectric surface, where the influence of the van der Waals force is negligible. The light intensities (see Table 2) are chosen so that the trap depth is 100 μK. The sum of all light intensities gives the total atomic light scattering rate at the center of the trap, $\Gamma_{\mathrm{sc}}=8$ Hz by Equation (Equation 4).

The extent of the lateral localization can be easily controlled by varying the intensity of the $\lambda_{3}=633$ nm (TE01) lightwave, as shown in Figure 4D, where cross-sections of the potential well at x=665 nm are shown. The dotted cyan line corresponds to the absence of the auxiliary lateral localization (i.e., this is the horizontal cross-section of Figure 4A). The solid magenta line is the horizontal cross-section of Figure 4B, while the other two lines show how the lateral confinement increases as the intensity of the TE01 auxiliary lightwave increases.

### 3.3. Techniques to Mitigate Temperature Rise Caused by Optical Absorption in Waveguides

Since the high intensity of the optical fields (used to trap the atoms) can potentially heat up and even damage the optical waveguides, possible ways to mitigate this problem should be mentioned. First, it should be noted that in waveguides on planar 1D PC, these problems are somewhat reduced by default, since the intensity maximum is near the interface and about half of the Poynting energy is transmitted through the vacuum. In a standard waveguide, only a small fraction of the total intensity is used to trap atoms, while the core of the waveguide is heated by the main part of the optical intensity.

A similar problem routinely arises in the context of the trapping of micrometer-sized particles in water by surface nanostructures using near-field optical tweezers [52,53,54,55]. There, even a moderate increase in the temperature of the nanostructures leads to convection of the water (in which the trapping takes place) and even to boiling, which prevents the trapping. To avoid this, several techniques have been proposed, including integrating a heat sink feature into the structure or using a substrate with high thermal conductivity. The same approaches can be used for the waveguide on planar 1D PC if optical absorption within the waveguide becomes an issue. The main problem with heat dissipation is the relatively low thermal conductivity of glasses and other transparent materials at room temperature, e.g., the thermal conductivity of pure fused silica is $\kappa_{{\mathrm{SiO}}_{2}}=1.38\;W/\left(m\cdot K\right)$ [56,57]. The use of sapphire ($\kappa_{\mathrm{sapphire}}=30\;W/\left(m\cdot K\right)$ [58]) as a substrate or indium tin oxide (ITO, $\kappa_{\mathrm{ITO}}=10.2\;W/\left(m\cdot K\right)$ [59]) as an integrating sublayer is one of the possible ways to distribute heat from the waveguide width in the lateral direction. Another way is to deposit gold films ($\kappa_{\mathrm{Au}}$∼250–300W/(m·K), depending on the film thickness [60]), parallel to the waveguide at a distance of about 10μm from its left and right sides, where the waveguide lateral optical field vanishes.

In contrast to the experiments with optical tweezers in water, the allowable temperature rise in our case is much higher and is limited only by structural deformation and degradation of the glass waveguide. The typical glass transformation temperature $T_{g}$ is above 500 °C [61].

### 3.4. Excitation of Waveguide Modes in a Kretschmann-like Scheme

For the excitation of only two/three necessary modes and for avoiding the excitation of undesired additional modes (e.g., ρ=1.17 at $\lambda_{2}=640$ nm on a flat surface; see Figure 2B, or ρ=1.016 at $\lambda_{3}=633$ nm (TE00) in the rib waveguide), the Kretschmann-type scheme is quite convenient. In this scheme, mode excitation takes place via a coupling prism on which the optimal number of double layers *N* (see (Equation 2)) is deposited. For the presented photonic structure, this number for the optimal excitation is N≃5. With optimal excitation of a surface wave, there is no reflection of an incident beam due to destructive interference of the reflected wave with the optical surface wave reradiated back to the prism. Only modes with $n_{\mathrm{eff}}$ close to unity can be excited if the beam excitation angle $\theta_{0}$ (see Figure 1) is chosen near the total internal angle (i.e., near ρ∼1). The prism coating can be arranged in such a way that half of the prism surface is covered by a PC structure (Equation 2) with N≃5 (for excitation), while the other half has the same structure with N>10 (for propagation without reradiation back into the prism).

## 4. Discussion

Engineering the optimal interface for the light–atom interaction is critically important for developing compact and robust quantum sensors based on guided atom interferometers. In this communication, a novel waveguide structure based on planar 1D PC is presented, which is designed to trap and waveguide neutral atoms by optical dipole force. This photonic structure makes it possible to increase penetration depths of the evanescent light waves, which provides very stable trapping of atoms at relatively large distances from the dielectric surface of the waveguide. Additional stabilization of the atom in the lateral dimension by the auxiliary TE01 mode was also considered. By modulating the intensity of this mode, the lateral width of the well trap can be easily modulated, which can be used, e.g., for adiabatic cooling and other manipulations of ultracold atomic ensembles.

## Figures and Tables

**Figure 1 sensors-23-08812-f001:**
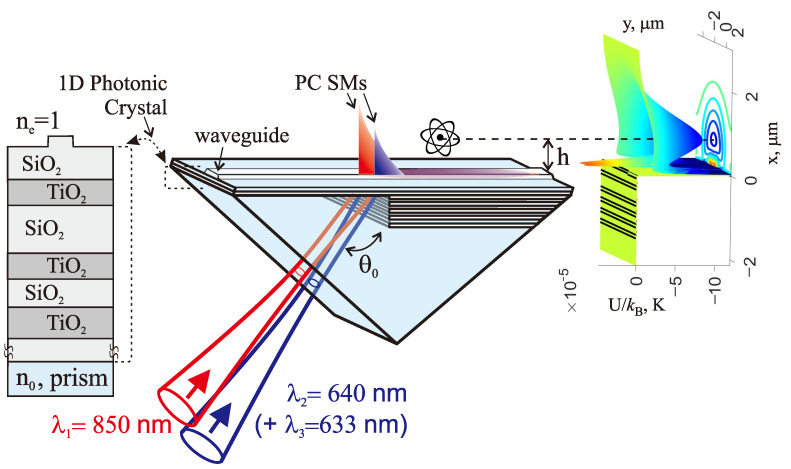
Concept of the photonic system under consideration: the left inset shows the cross-section of the 1D PC structure, and the right inset shows the outline of the well trap potential for ultracold atoms.

**Figure 2 sensors-23-08812-f002:**
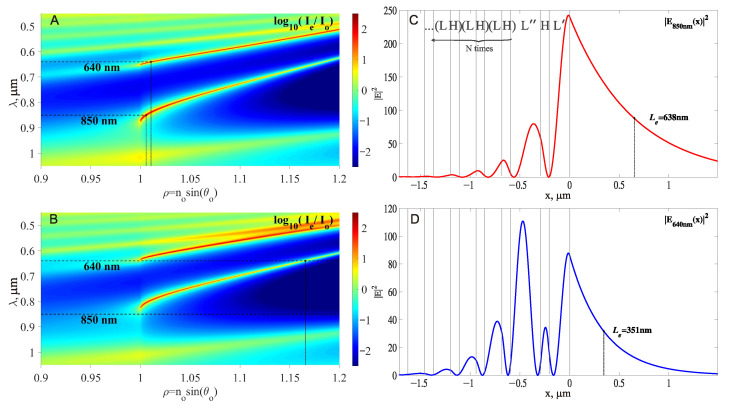
Dispersion of the planar 1D PC structure (Equation 2), for N=5, with $L^{\prime }$ thickness $d_{3}=207$ nm (**A**) and with $d_{3}=182$ nm (**B**). Intensity profiles of modes at $d_{3}=207$ nm for $\lambda_{1}=850$ nm, $\rho =n_{\mathrm{eff}}\left(w_{\infty }\right)=1.0056$ (**C**) and $\lambda_{2}=640$ nm, $\rho =n_{\mathrm{eff}}\left(w_{\infty }\right)=1.0105$ (**D**). The structural formula in use (Equation 2) is displayed at the top of (**C**).

**Figure 3 sensors-23-08812-f003:**
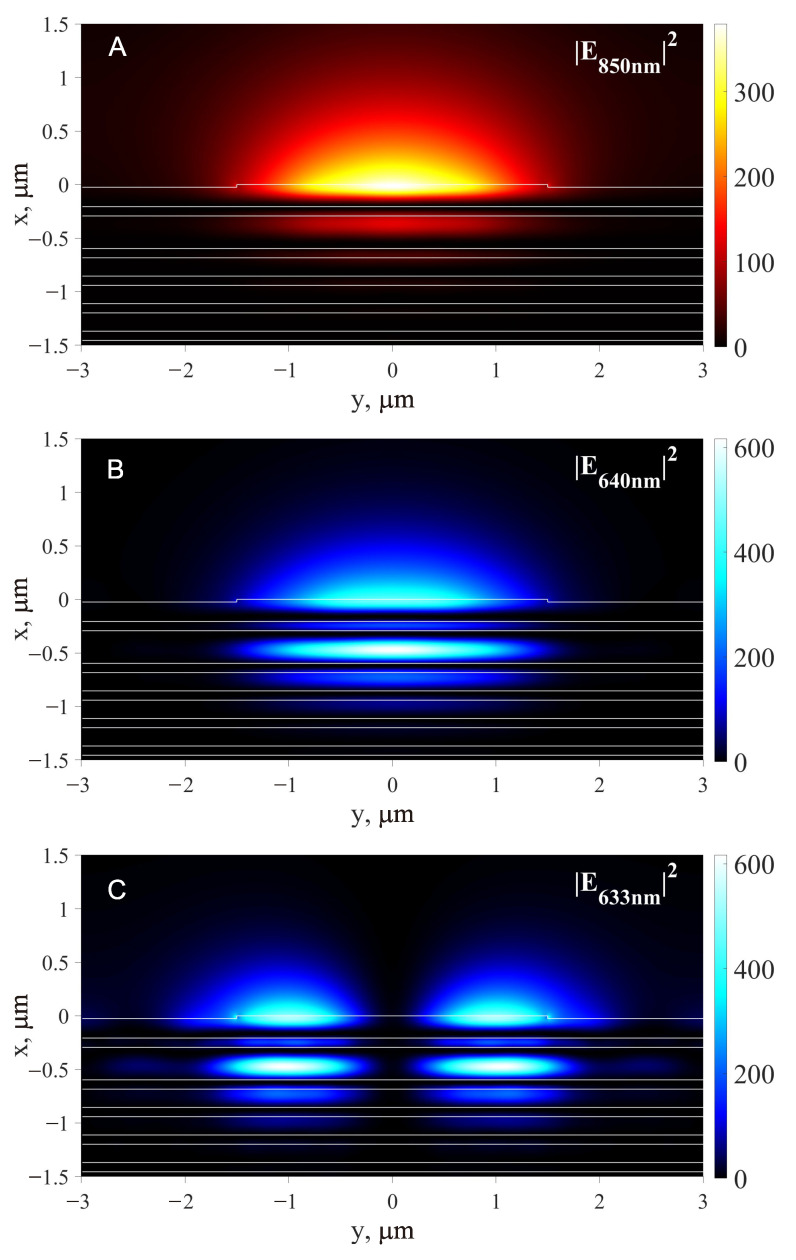
Electric field intensity distribution for red-detuned attracting light beam, $\lambda_{1}=850$ nm, $n_{{\mathrm{eff}}_{1}}=1.0012$ (**A**), blue-detuned repulsive light beam, $\lambda_{2}=640$ nm, $n_{{\mathrm{eff}}_{2}}=1.0074$ (**B**) and auxiliary blue-detuned repulsive light beam for lateral support, $\lambda_{3}=633$ nm, $n_{{\mathrm{eff}}_{3}}=1.0076$ (**C**).

**Figure 4 sensors-23-08812-f004:**
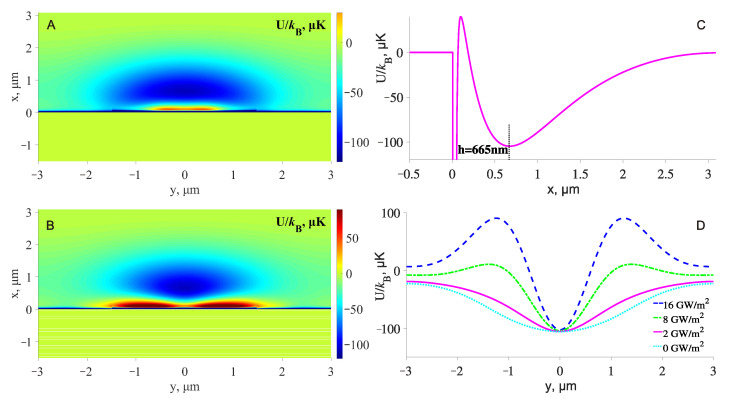
Optical potential well when two wavelengths of 850 nm and 640 nm are involved (**A**), and when the third auxiliary wavelength of 633 nm is added (**B**). Vertical cross-section of the potential well at y = 0 (**C**) and horizontal cross-sections of the potential well at x = 665 nm for different intensities of the 633 nm wave (**D**).

**Table 1 sensors-23-08812-t001:** The RIs of the dielectric layers and the PC SMs parameters for the planar 1D PC.

Wavelength	$n_{1}{\left({\mathrm{SiO}}_{2}\right)}^{⋆}$	$n_{2}{\left({\mathrm{TiO}}_{2}\right)}^{⋆}$	$\rho =n_{\mathrm{eff}}\left(w_{\infty }\right)$	$L_{e}$, nm
850 nm	1.4666	2.3137	1.0056	638
640 nm	1.4679	2.3674	1.0105	351

^⋆^ + *i* × $10^{-5}$.

**Table 2 sensors-23-08812-t002:** The calculated mode parameters for the rib waveguide with w=3μm,h=25nm.

Wavelength	$n_{\mathrm{eff}}\left(w,h\right)$	Vacuum Fraction, $S_{z}^{e}/S_{z}^{\mathrm{total}}$	Mode Effective Area, $\mu m^{2}$	Loss ^⋆^, dB/cm	Optical Power in Figure 4, mW	Intensity in Figure 4, GW/m^2^
850 nm	1.0012 ^†^	76%	6.0	2	8.4	1.4
640 nm	1.0074 ^†^	46%	3.5	7	29.1	8.3
633 nm	1.0076 ^‡^	39%	3.7	8	7.4	2.0

^†^ TE00; ^‡^ TE01; ^⋆^ for $Im\left(n_{j}\right)=10^{-5},\forall j$.

## Data Availability

The data that support the findings of this study are available from the corresponding author upon reasonable request.

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
