# Peer review of "Photonic Crystal Surface Modes for Trapping and Waveguiding of Ultracold Atoms"

_sensors, 2023, doi:10.3390/s23218812_

Round 1

Reviewer 1 Report

Comments and Suggestions for Authors

In this paper, the author provides a novel way to construct compact quantum sensor. However, the detailed experimental data and analysis are not enough. Therefore, this paper should be revised before published. Here are some suggestions.

1.     The quantitative improvement of system should be added in the abstract.

2.     The main innovation of this paper is not clear. The author should add some text to explain the innovations in the end of Introduction.

3.     In Figure 2, the text is too small. And the unit of color axis is not explicit.

4.     The experiment of paper is too simple. The author should add additional experiment to show the performance of system. For example, the quantum sensor using by the waveguiding is testing.

5.     In general, the article lacks quantitative analysis. For example, the experimental results in Fig.3 and Fig. 4 should be explained further.

6.     A few relating works are suggested to reference,

“Numerical study of biosensor based on α-MoO3/Au hyperbolic metamaterial at visible frequencies”. Journal of Physics D: Applied Physics, 2020, 54(3): 034001.

“Plasmonic anapole metamaterial for refractive index sensing”. PhotoniX 3, 23 (2022).

A Reversible Tuning of High Absorption in Chalcogenide–Metal Stacked-Layer Structure and Its Application for Multichannel Biosensing. Advanced Photonics Research, 2021, 2(8): 2000152.

Liquid crystal-amplified optofluidic biosensor for ultra-highly sensitive and stable protein assay. PhotoniX 2, 18 (2021).

Comments on the Quality of English Language

The grammar needs to be further polished.

Author Response

see Response.pdf file

Reviewer 2 Report

Comments and Suggestions for Authors

The article describes the design and simulation of an optical system for trapping cold atoms. It is a purely theoretical article, yet it is well-written and well-structured. The reading is quite straightforward even for those who are not experts in the field. Overall, I would say that the interest in this type of article can be considered moderate to low since it is only a simulation. However, it should be emphasized that the proposed reasoning is entirely consistent, and most likely, if an experimental group were to attempt to build the device, they could achieve results consistent with the simulation. On the other hand, it is also true that the design of an optical structure using a simulator could lead to a multitude of seemingly original and potentially functional ideas. Overall, I believe that the article can still be published. I suggest only a few minor revisions. In particular, I think that the readability would be improved if a diagram were used instead of (2). This would address a potential interpretation difficulty for a somewhat distracted reader who may not understand what the author is referring to when discussing the number of double layers in paragraph 3.3.

For the sake of clarity and comprehensiveness, it's worth emphasizing that the presence of an evanescent field in a dielectric photonic crystal structure, such as a Bragg mirror, is not a novelty. Therefore, the only novelty in the article would be the engineering of the system. For this reason, the manuscript is primarily of a technical nature.

Author Response

see Response.pdf file

Reviewer 3 Report

Comments and Suggestions for Authors

The Authors propose the design of a photonic system for trapping and waveguiding of ultracold atoms above a dielectric surface. The reported results have been achieved by simulations. The manuscript is very interesting. Therefore, it deserves the publications after addressing the following comments.

- The Author has to rate other trapping technologies reported at the state of the art (see, i.e., Nanoscale optical trapping by means of dielectric bowtie. In Photonics (Vol. 9, No. 6, p. 425). MDPI, 2022; Trapping-assisted sensing of particles and proteins using on-chip optical microcavities. ACS nano7(2), 1725-1730, 2013; Optical trapping of nanoparticles using all-silicon nanoantennas. ACS Photonics5(12), 4993-5001, 2018).

- The simulation approach has to be widely discussed.

- It is not clear how the trapped particles moves along the waveguide. Moreover, since the optical input power is so large, the temperature effects has to be simulated. Large input power leads to a strong increase of temperature, that in turn affect the stability of trapping.

Author Response

see Response.pdf file

Round 2

Reviewer 3 Report

Comments and Suggestions for Authors

In the reply to the question #1, the Author reports that the bibliography on trapping has been improved, however it doesn't appear to be done. 

Author Response

In the revised version of the manuscript, two references [52,53] have been added . In the current revised_2 version, ALL references provided by Reviewer #3 have been added. These are references [52-55] in the new numbering.

I hope this improves the manuscript quality.

Valery Konopsky.

PS. to the Editor:
Please note that the manuscript type has been changed from "Articles" to "Communication", as I indicated in my previous Response.